# Improving Sampling Efficiency in RLVR through Adaptive Rollout and Response Reuse

## Abstract

Large language models (LLMs) have achieved impressive reasoning performance, with reinforcement learning with verifiable rewards (RLVR) emerging as a standard paradigm for post-training. A representative algorithm, group relative policy optimization (GRPO) (Shao et al., 2024), computes advantages by normalizing outcome rewards within response groups, but suffers from a vanishing advantage issue when all responses in a group receive identical rewards. To address this issue, we propose Adaptive Rollout and Response Reuse Policy Optimization (AR3PO), a sampling efficient RLVR algorithm that introduces two novel techniques: *adaptive rollout*, which dynamically allocates more responses to difficult prompts while saving computation on easier ones, and *response reuse*, which leverages previously generated correct responses to provide useful training signals. We compare AR3PO with strong RLVR baselines on multiple representative benchmarks using two different families of base models. Across the 7B and 8B models, AR3PO consistently outperforms GRPO and matches or surpasses DAPO (Yu et al., 2025), reducing rollout cost by up to 4.2×. On the larger 32B model, AR3PO achieves comparable performance to DAPO at similar training steps while maintaining substantially lower rollout cost.

## 1 Introduction

Large language models (LLMs) have demonstrated remarkable reasoning capabilities across diverse domains, including mathematics (Hendrycks et al., 2021), coding (Chen et al., 2021), and scientific problem solving (Rein et al., 2024). *Reinforcement Learning with Verifiable Rewards* (RLVR) has played a central role in this progress (Jaech et al., 2024; Team et al., 2025; Guo et al., 2025), and has emerged as a standard paradigm for post-training LLMs on reasoning tasks. In RLVR, a verifier is used to provide rule-based outcome rewards. For instance, in mathematical tasks, the reward is a binary indicator of whether the generated response is correct. Policy gradient algorithms are commonly employed in RLVR to train LLMs, with group relative policy optimization (GRPO) (Shao et al., 2024) being a representative example. GRPO builds on the proximal policy optimization (PPO) update (Schulman et al., 2017), but a key distinction is that GRPO does not require training a separate value network to estimate advantages. Instead, it computes advantages through group normalization: for each prompt, GRPO generates a group of responses and normalizes their outcome rewards within the group to obtain the advantages. This design improves training stability and has demonstrated strong performance in DeepSeek-R1 (Guo et al., 2025).

However, GRPO faces a limitation in its normalized advantage computation: when all responses within a group are either correct or incorrect, the verifier assigns identical rewards to all responses, causing the advantages to collapse to zero and yielding no training gradients. To address this vanishing advantage issue, DAPO (Yu et al., 2025) introduces a dynamic sampling strategy that continues sampling new prompts and responses until every group contains non-zero reward variance. While this approach alleviates the problem, it incurs substantially higher computational costs for response generation. According to their official implementation, DAPO requires at least three times more response generation than standard GRPO, which quickly becomes a computational bottleneck for large models. This motivates us to study the following question:

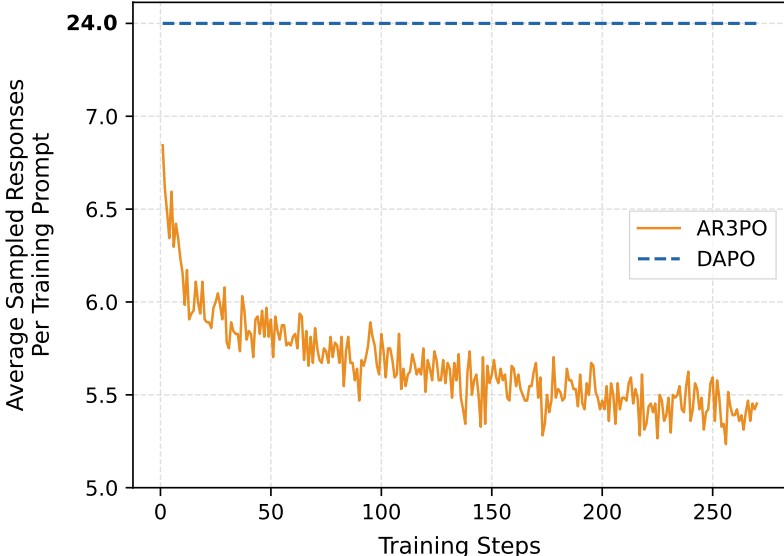

Figure 1: Comparison of the average number of sampled responses per training prompt between DAPO and our AR3PO algorithm. By leveraging our proposed adaptive rollout and response reuse techniques, AR3PO requires fewer responses as training progresses, with a final average of 5.7, reducing generation cost by approximately 4.2× compared to DAPO.

*How can we address the vanishing advantage issue in a more sampling efficient way?*

Our proposed solution builds on two key observations about GRPO's potential inefficiency:

1. GRPO generates a fixed number of responses for each prompt, regardless of its difficulty. This uniform allocation can be suboptimal: harder prompts often require more responses to produce at least one correct answer and yield training gradients, whereas easier prompts may not need as many. Moreover, as training progresses, the model may generate only correct responses for easy prompts, resulting in a waste of the rollout computation.

2. GRPO only utilizes on-policy responses sampled from the current model, while discarding responses generated in earlier steps. As a result, for difficult prompts, all responses in the current step may be incorrect, yielding no training signal, even though a correct response was sampled previously but discarded.

**Contributions.** Based on these observations, we propose a novel algorithm AR3PO that combines two complementary ideas: *adaptive rollout* and *response reuse*. In adaptive rollout, the response generation process is divided into multiple stages, and only prompts without any correct response proceed to the next stage. This design allocates more budget to difficult prompts, increasing the likelihood of obtaining at least one correct response, while saving computation on easy prompts where correct responses can be generated with high probability. For prompts without any correct response after rollout, we propose reusing correct responses generated in earlier steps. However, the behavior policy that generated these responses may differ substantially from the current policy, which can lead to importance ratios that are either excessively small or large. To address this, we introduce two new techniques: (1) reusing only the reward information of the past correct response in the advantage computation and performing training on the on-policy samples; (2) recomputing the token probabilities under the current policy to reduce the variance of the objective.

We compare AR3PO with GRPO and DAPO on mathematical reasoning tasks. Across two different base models Qwen2.5-7B (Qwen, 2024) and Llama-3.1-8B-Instruct (Dubey et al., 2024), AR3PO consistently outperforms GRPO and achieves performance comparable to or slightly better than DAPO. More importantly,

AR3PO significantly improves sampling efficiency, reducing generation cost up to 4.2× compared to DAPO, as illustrated by the Qwen results in Figure 1. On the larger Qwen2.5-32B model, AR3PO achieves performance comparable to DAPO at similar training steps, again with substantially lower rollout cost[1]. Our study also validate the effectiveness of the two proposed techniques: *adaptive rollout* conserves generation budget on easy prompts while allocating more responses to difficult ones, whereas *response reuse* reduces the proportion of prompts without any correct response from about 0.3 to 0.2. Overall, our results establish AR3PO as a sampling efficient and effective approach for RLVR.

## 2 Preliminaries

**Notations.** We denote a prompt by $x \in \mathcal{X}$, where $\mathcal{X}$ is the prompt space. An LLM is characterized by a policy $\pi_\theta : \mathcal{X} \to \Delta(\mathcal{O})$ that maps a prompt to a distribution over the output space $\mathcal{O}$.

### 2.1 Group Relative Policy Optimization (GRPO)

GRPO (Shao et al., 2024) is the post-training algorithm employed in DeepSeek-R1 (Guo et al., 2025) to enhance reasoning performance. At each step, a batch of question–answer pairs $(x, a)$ is sampled from the dataset $\mathcal{D}$, and the current policy $\pi_\theta$ generates a group of responses $\{o_i\}_{i=1}^G$ for question $x$.[2] For each response $o_i$, a math verifier provides a binary reward $R_i$ according to the answer $a$, and the advantage $A_i$ is computed by normalizing rewards within the group:

$$A_i = \frac{R_i - \text{mean}(\{R_i\}_{i=1}^G)}{\text{std}(\{R_i\}_{i=1}^G)}. \tag{1}$$

The GRPO objective is then defined as:

$$
\begin{aligned}
\mathcal{J}_{\text{GRPO}}(\theta') = & \mathbb{E}_{(x,a)\sim\mathcal{D},\{o_i\}_{i=1}^G \sim \pi_\theta(\cdot|x)} \\
& \left[ \frac{1}{\sum_{i=1}^G |o_i|} \sum_{i=1}^G \sum_{t=1}^{|o_i|} \min\left( r_{i,t}(\theta') A_i, \text{clip}\left( r_{i,t}(\theta'), 1 - \epsilon_{\text{low}}, 1 + \epsilon_{\text{high}} \right) A_i \right) \right],
\end{aligned} \tag{2}
$$

where $r_{i,t}(\theta')$ is the importance ratio:

$$r_{i,t}(\theta') = \frac{\pi_{\theta'}(o_{i,t} \mid x, o_{i,<t})}{\pi_\theta(o_{i,t} \mid x, o_{i,<t})}. \tag{3}$$

Here we adopt the token-level GRPO objective and remove the KL penalty term, following the recommendation of Yu et al. (2025) for improved training. Compared to PPO (Schulman et al., 2017), GRPO eliminates the need for a value function estimator, thereby stabilizing training and improving efficiency. However, in the RLVR setting, it is possible that all the responses within a group are either correct or incorrect, particularly when the question is too easy or too hard for the current model. In such cases, all advantages $A_i$ become zero, contributing no gradient signals for training.

### 2.2 Dynamic Sampling Policy Optimization (DAPO)

To address the issue of vanishing advantages, Yu et al. (2025) propose a dynamic sampling strategy that repeatedly draws new questions until the generated responses within a group are not uniformly correct or incorrect, thereby ensuring non-zero reward variance for training. The corresponding objective can be

---

[1] See Section 4.5 for details.

[2] In this paper, prompts correspond to mathematical questions fed to the LLM, and we use the terms prompt and question interchangeably.

formulated as:

$$\mathcal{J}_{\text{DAPO}}(\theta') = \mathbb{E}_{(x,a)\sim\mathcal{D},\{o_i\}_{i=1}^{G}\sim\pi_\theta(\cdot|x)}$$

$$\left[ \frac{1}{\sum_{i=1}^{G}|o_i|} \sum_{i=1}^{G} \sum_{t=1}^{|o_i|} \min\left(r_{i,t}(\theta')A_i, \text{clip}\left(r_{i,t}(\theta'), 1-\epsilon_{\text{low}}, 1+\epsilon_{\text{high}}\right)A_i\right) \right]$$

$$\text{s.t.} \quad 0 < \left|\left\{o_i \mid \text{is\_equivalent}(a, o_i)\right\}\right| < G.$$

Although this approach mitigates the vanishing advantage problem, it introduces additional computational overhead due to repeated generations, which becomes particularly costly for larger models.

## 3 Algorithm

In this section, we propose a novel algorithm AR3PO that mitigates the vanishing advantage issue in a more sampling efficient manner. We improve GRPO along two dimensions: (1) a multi-stage rollout process that allocates more responses to difficult prompts while saving generation resources on easy prompts where correct responses can be obtained reliably; and (2) reusing correct responses from earlier steps for prompts that fail to yield any correct response in the current step. These two improvements correspond to our two newly proposed techniques, *adaptive rollout* and *response reuse*.

### 3.1 Adaptive Rollout

To improve GRPO from the first aspect, we propose a new adaptive rollout strategy for response generation. In each training step, we divide the rollout process into $S$ stages and maintain a prompt pool $\mathcal{U}$. At stage $s$, the model generates $k$ responses for each $q \in \mathcal{U}$. The prompt with at least one correct response are removed from $\mathcal{U}$ and the process is repeated until $\mathcal{U}$ becomes empty or the maximal generation step is reached. Compared to uniform response generation, adaptive rollout has two advantages:

- For easy prompts where the model has a high probability of generating correct responses, our strategy generates fewer responses. This not only reserves more rollout budget for difficult prompts but also mitigates the issue where all responses are correct.

- For difficult prompts, the strategy allocates more rollout budget, increasing the probability of obtaining at least one correct response and thus yielding a non-zero advantage.

Our method can also be viewed as a form of adaptive weighting over prompts. Since each generated response serves as a training sample for updating the model, allocating more responses to difficult prompts effectively increases their weight in the optimization, while correspondingly reducing the weight of easier prompts.

Even after adaptive rollout, some prompts may still yield no correct responses. This limitation motivates our second technique, *response reuse*.

### 3.2 Response Reuse

In PPO-style algorithms such as GRPO and DAPO, only on-policy samples are used for model updates, while previously generated off-policy samples are discarded. This data inefficiency poses particular challenges in the RLVR setting: when the model fails to produce a single correct response within a group, it receives no training signal for updates, even though correct responses to the same prompt may have been generated in earlier steps. This motivates us to reuse previously generated responses, with a particular emphasis on correct ones.

Specifically, we maintain a replay buffer $\mathcal{B}$ that stores all previously generated correct responses. For prompts without any correct response after the adaptive rollout process, we randomly select one response $o_c$ from $\mathcal{B}$ to replace an incorrect response in the group. Without loss of generality, we assume $o_G$ is replaced by $o_c$, and the advantages for the updated group are then recomputed as in Eq. 1.

---

**Algorithm 1** Adaptive Rollout and Response Reuse Policy Optimization (AR3PO)

---

1: Initialize policy $\pi_\theta$, task prompts $\mathcal{D}$, replay buffer $\mathcal{B} \leftarrow \emptyset$
2: **for** step $= 1, \cdots, T$ **do**
3:      Sample a prompt batch $\mathcal{D}_b$ from $\mathcal{D}$
4:      $\mathcal{U} \leftarrow \mathcal{D}_b$
5:      **for** $s = 1, \cdots, S$ **do**
6:          Generate $k$ responses $\{o_i\}_{i=1}^k \sim \pi_\theta(\cdot \mid q)$ for each $q \in \mathcal{U}$
7:          Obtain binary rewards $\{R_i\}_{i=1}^k$ via a math verifier
8:          Remove prompts with at least one correct response from $\mathcal{U}$
9:      **end for**
10:      For remaining $q \in \mathcal{U}$, replace one incorrect response with a correct response randomly sampled from $\mathcal{B}$ if available
11:      Compute normalized advantages for all responses with Eq. 1
12:      **if** using off-policy learning **then**
13:          Recompute the token probabilities of reused responses with $\pi_\theta$
14:      **else**
15:          Stop gradient on reused responses
16:      **end if**
17:      Update $\pi_\theta$ by gradient ascent on Eq. 2
18:      Update replay buffer $\mathcal{B}$ with new responses where $R_i = 1$
19: **end for**

---

As a result, the responses within the group are sampled from two different policies. For $\{o_i\}_{i=1}^{G-1}$, they are sampled from the current policy $\pi_\theta$, and their importance ratios are computed as in Eq. 3. In contrast, the reused response $o_c$ is generated by a previous policy $\pi_{\theta_{\text{old}}}$, and its corresponding importance ratio is given by:

$$r_{c,t}(\theta') = \frac{\pi_{\theta'}(o_{c,t} \mid x, o_{c,<t})}{\pi_{\theta_{\text{old}}}(o_{c,t} \mid x, o_{c,<t})}.$$

However, $\pi_{\theta_{\text{old}}}$ may differ substantially from the current policy, leading to importance ratios that are either excessively large or vanishingly small. Large ratios increase the variance and destabilize training, while small ratios diminish gradient magnitudes during updates. To address this issue, we propose two new techniques:

1. Use the current policy $\pi_\theta$ as the behavior policy in $r_{c,t}(\theta')$, i.e., replace the denominator of $r_{c,t}(\theta')$ with $\pi_\theta(o_{c,t} \mid x, o_{c,<t})$. In practice, this corresponds to recalculating the token probabilities of $o_c$ under the current policy $\pi_\theta$. Although this modification introduces bias into the optimization objective, it is well established in reinforcement learning that controlling the variance of policy gradient estimates is often more critical than reducing bias (Sutton et al., 1998).

2. Stop the gradient on $o_c$ and update the model only using the on-policy samples $\{o_i\}_{i=1}^{G-1}$. This can be interpreted as a form of negative sample training: since a previous policy was already able to generate correct responses for this prompt, the current model should have already acquired the knowledge required to solve it during pre-training. Therefore, although the current model fails to generate a correct response in this step, we reuse the reward information from previous responses and assign negative advantages to the incorrect responses, thereby discouraging the policy from exploring wrong directions.

### 3.3 Adaptive Rollout and Response Reuse Policy Optimization (AR3PO)

Combining the two techniques described above, we propose a new algorithm, *adaptive rollout and response reuse policy optimization* (AR3PO), summarized in Algorithm 1. After the adaptive rollout stage, AR3PO offers two options: (i) perform off-policy learning on the reused responses with updated token probabilities, or (ii) stop the gradient on the reused response and update the model only on the on-policy samples. The latter option is computationally more efficient, as it avoids gradient computation on the off-policy sample.

Recent works have also studied how to allocate rollout budgets more effectively than uniform allocation (Yao et al., 2025; Liao et al., 2025). In contrast, our adaptive rollout is not aimed at maximizing performance under a fixed generation budget. Rather, it is designed to improve sampling efficiency and reduce the computational cost of generation, while still ensuring informative training signals. Compared to concurrent works that directly apply rollout replay (Sun et al., 2025; Zhang et al., 2025a), our method further introduces new techniques to mitigate issues arising in the importance ratio term.

## 4    Experiments

### 4.1    Experimental Setup

**Datasets and Models.**    We focus on mathematical reasoning tasks and adopt the DAPO-Math dataset (Yu et al., 2025) as our training set. The original dataset contains 17K prompts, each associated with an integer answer. After removing non-English prompts, 14K prompts remain. We evaluate model performance on four representative mathematical benchmarks: Math500 (Hendrycks et al., 2021), Minerva Math (Lewkowycz et al., 2022), Olympiad Bench (He et al., 2024), and AIME 2024[3]. OlympiadBench includes multimodal questions in both mathematics and physics; we restrict our evaluation to the text-only mathematics subset. For Math500, Minerva Math, and OlympiadBench, we report `avg@8`, as these datasets contain hundreds of problems. For AIME 2024, which contains only 30 problems, we report `avg@64`. We use Math-Verify[4] as the verifier to check the correctness of the output answer. To better assess the generality of our algorithms, we conduct experiments on base models from different families: Qwen2.5-7B (Qwen, 2024) and Llama-3.1-8B-Instruct (Dubey et al., 2024).

**Implementation Details.**    We compare AR3PO against two strong RLVR baselines: GRPO (Shao et al., 2024) and DAPO (Yu et al., 2025). All algorithms are implemented using the VERL framework (Sheng et al., 2025). For GRPO and DAPO, we generate 8 responses per prompt, while AR3PO adopts $S = 2$ and $k = 4$, resulting in at most 8 responses per prompt. We use a training prompt batch size of 512 and a mini-batch size of 128 for gradient updates. For DAPO, we follow its official implementation of the dynamic sampling strategy, which employs a data prompt batch size of 1536.

For Qwen2.5-7B, the maximum prompt length is set to 1024 and the maximum response length to 3072, with a learning rate of $1 \times 10^{-6}$. For Llama-3.1-8B-Instruct, the maximum prompt length is also set to 1024 and the maximum response length to 2048. Since this model has already undergone RLHF post-training, we adopt a smaller learning rate of $1 \times 10^{-7}$.

For both models, we do not apply learning rate warmup and follow the default hyperparameter settings in the VERL framework for all other configurations. All algorithms are trained under the same hyperparameter settings to ensure fair comparison.

### 4.2    Main Results

We implement AR3PO with the second option, which incorporates the reused response in the advantage computation while updating the model on on-policy samples. We compare this implementation against two baselines on four mathematical benchmarks, with results reported in Table 1. AR3PO consistently outperforms GRPO and achieves performance comparable to or slightly better than DAPO. More importantly, AR3PO significantly improves sampling efficiency, reducing generation cost by 4.2× and 3.6× compared to DAPO. This improvement is mainly attributed to the *adaptive rollout* technique, which saves budget on easy prompts and allocates more responses to difficult ones. In addition, the *response reuse* technique ensures that even when no correct response is generated in the current step, the model can still obtain meaningful training signals from difficult prompts, thereby enhancing overall performance.

---

[3]https://huggingface.co/datasets/Maxwell-Jia/AIME_2024
[4]https://github.com/huggingface/Math-Verify

Table 1: Comparison of AR3PO with baselines on four mathematical benchmarks. The penultimate column reports the average number of sampled responses per training step (computed as generation batch size × number of rollouts), and the last column shows the rollout speedup relative to DAPO.

| Model | Method | Math 500 | Minerva Math | Olympiad Bench | AIME 24 | Average | Sampled Responses | Speedup (vs. DAPO) |
|-------|--------|----------|--------------|----------------|---------|---------|-------------------|---------------------|
| Qwen | GRPO | 77.5 | 37.4 | 38.8 | 15.2 | 42.2 | 512 × 8.0 | 3.0 |
| | DAPO | 77.2 | 36.4 | 41.1 | 16.9 | 42.9 | 1536 × 8.0 | 1.0 |
| | AR3PO | 78.8 | 36.0 | 39.6 | 18.0 | **43.1** | 512 × 5.7 | **4.2** |
| Llama | GRPO | 52.6 | 26.6 | 19.9 | 6.7 | 26.5 | 512 × 8.0 | 3.0 |
| | DAPO | 53.6 | 28.1 | 20.3 | 9.2 | 27.8 | 1536 × 8.0 | 1.0 |
| | AR3PO | 53.7 | 26.6 | 21.2 | 9.5 | **27.8** | 512 × 6.7 | **3.6** |

Table 2: Comparison of different response reuse strategies using Qwen2.5-7B as the base model. Direct rollout replay uses token probabilities from the previous behavior policy in the importance ratio. Option I applies our first technique, recomputing token probabilities with the current policy. Option II applies our second technique, incorporating reused responses in the advantage computation while updating the model on on-policy samples.

| Method | Math 500 | Minerva Math | Olympiad Bench | AIME 24 | Average |
|--------|----------|--------------|----------------|---------|---------|
| AR3PO w/ direct rollout replay | 77.3 | 35.7 | 39.1 | 15.0 | 41.8 |
| AR3PO w/ option I | 76.9 | 35.8 | 39.3 | 19.3 | 42.8 |
| AR3PO w/ option II | 78.8 | 36.0 | 39.6 | 18.0 | **43.1** |

### 4.3 Analysis of Response Reuse Strategies

In this subsection, we compare different response reuse strategies, with results reported in Table 2. Direct rollout replay, which uses token probabilities from the behavior policy, performs the worst due to the potential discrepancy between the behavior policy and the current policy. This mismatch can lead to excessively small or large importance ratios, resulting in high variance in the objective. Our first technique recomputes token probabilities with the current policy, which introduces bias into the objective but reduces variance, thereby achieving performance comparable to DAPO. Our second technique, corresponding to the implementation in Table 1, performs the best, as it updates the model on on-policy samples while leveraging previous responses to compute advantages and provide useful training signals.

### 4.4 Training Dynamics

In this subsection, we present the dynamics of the training process. Figure 2a reports the average reward of generated responses at each step. Reward dynamics serve as an important monitoring metric in reinforcement learning, and we observe that the reward increases steadily throughout training. As training progresses and more responses are allocated to difficult prompts, the growth rate becomes slower compared to the initial phase. Figure 2b shows the average response length during training. We observe that the length increases rapidly at the beginning and then fluctuates around 1200. This observation is consistent with prior findings (Guo et al., 2025; Yu et al., 2025), which report that response length does not always increase monotonically and may even decrease during training. Since AR3PO allocates a varying number of responses across prompts, these two metrics are not directly comparable to those of GRPO and DAPO.

Figure 2c presents the ratio of prompts without any correct response for AR3PO and GRPO. The ratio curves nearly overlap during the first 30 steps, which correspond to the first training epoch. Afterward, as previously collected responses are reused, AR3PO reduces the ratio from about 0.3 to below 0.2, thereby providing more effective training signals for difficult prompts.

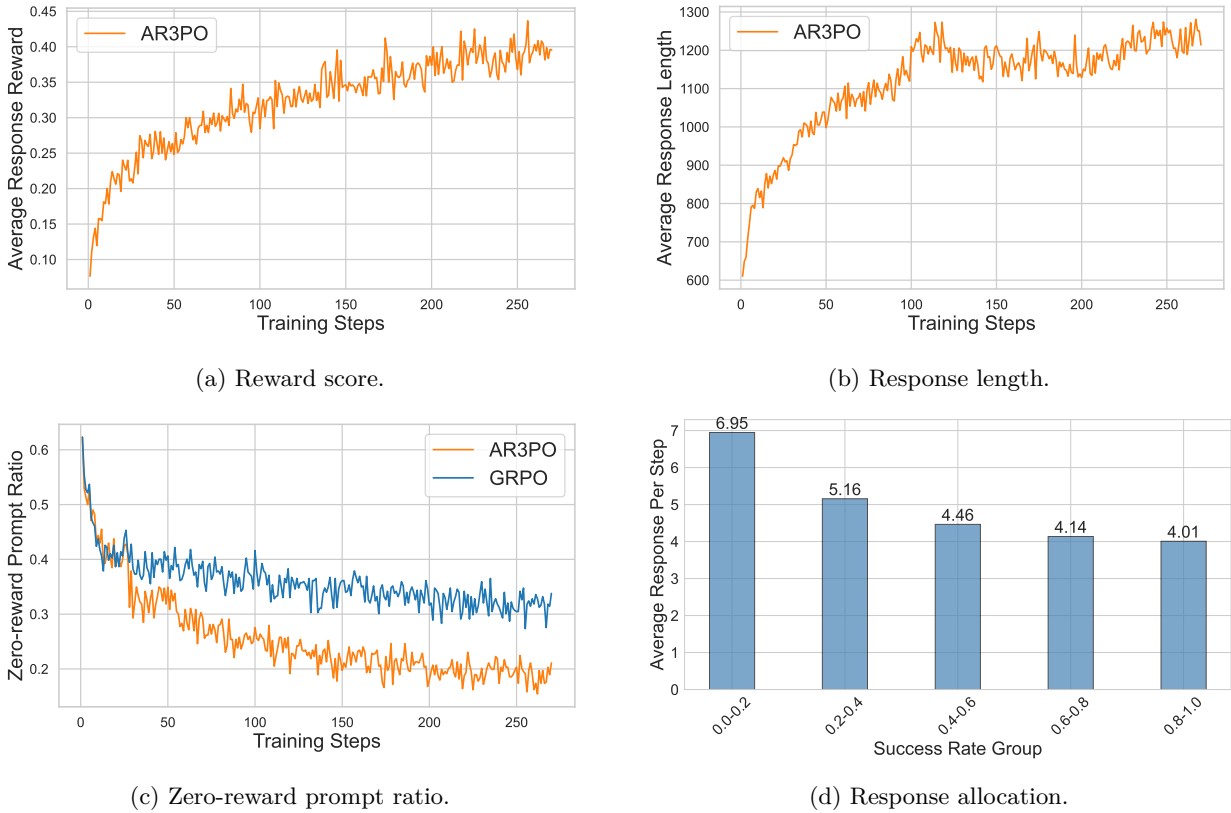

(a) Reward score.

(b) Response length.

(c) Zero-reward prompt ratio.

(d) Response allocation.

Figure 2: (a) and (b) show the metric curves of average reward and response length for AR3PO. (c) shows the ratio of prompts without any correct response in the group. (d) presents the average sampled responses per step for prompts grouped by cumulative success rate.

For prompts grouped by cumulative success rate, Figure 2d reports the average number of generated responses. The most difficult prompts with a success rate of 0.0-0.2 receive the largest allocation, averaging 6.95 responses, whereas the easiest prompts with a success rate of 0.8–1.0 use only about 4 responses per step. This demonstrates that our adaptive rollout strategy effectively saves generation budget on easy prompts and allocates it to difficult prompts, thereby improving the sampling efficiency.

## 4.5   Results on 32B Model

To further evaluate the generality of our algorithm on larger models, we follow the setup in Yu et al. (2025) and conduct experiments with the Qwen2.5-32B base model, as shown in Figure 3. We implement AR3PO with off-policy training, which shows better performance on difficult problems as demonstrated in the 7B model results. Due to computational constraints, we set the maximum response length to 4096 and train the model for 150 steps. AR3PO achieves performance comparable to DAPO at similar training steps while requiring only 5.3 responses per prompt with a generation batch size of 512. In contrast, DAPO uses 16 rollouts per prompt; since the paper does not report the number of prompts used in dynamic sampling, we cannot make a precise rollout cost comparison. As a reference, their later official implementation adopts a generation batch size of 1,536, under which AR3PO achieves a 9× reduction in rollout cost. Meanwhile, DAPO uses a maximum response length of 20480, whereas we limit it to 4096, suggesting that our method still has room for improvement with longer responses.

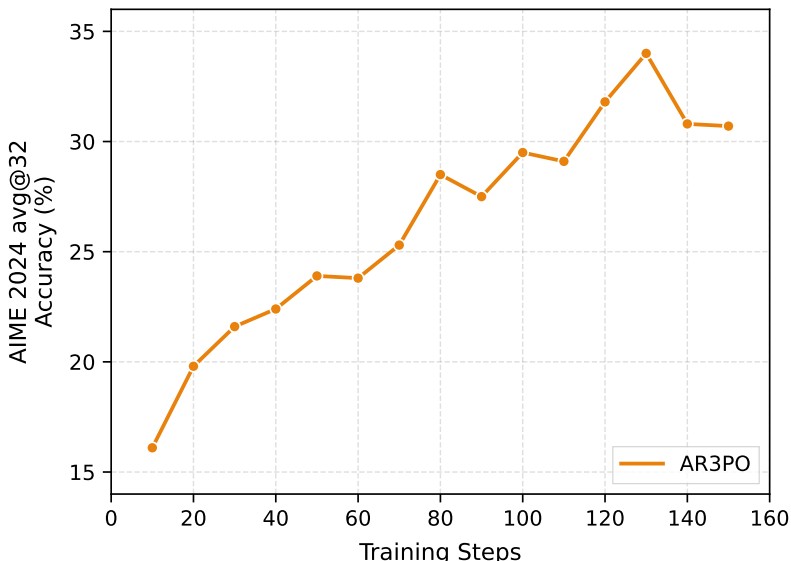

Figure 3: AIME 2024 accuracy of AR3PO with the Qwen2.5-32B base model. Our method achieves 34.0% accuracy at step 130, comparable to the performance reported by DAPO around 130 training steps (see Figure 1 in Yu et al. (2025)). Note that their figure reports gradient update steps, with 16 updates per step; thus, our step 130 approximately corresponds to their step 2000.

## 5 Related Work

**RL algorithms for LLM post-training.** Since the success of reinforcement learning from human feedback (RLHF) in ChatGPT (Ouyang et al., 2022; Achiam et al., 2023), reinforcement learning algorithms have been extensively explored for LLM post-training. To align LLMs with human preferences, Bai et al. (2022) employ the PPO algorithm (Schulman et al., 2017) to optimize a KL-regularized objective. Rafailov et al. (2024) propose direct preference optimization (DPO), which directly minimizes a loss function derived from the Bradley–Terry (BT) model to capture human preferences. Building on DPO, a number of variants have been developed, including offline algorithms such as KTO (Ethayarajh et al., 2024), ORPO (Hong et al., 2024), and SimPO (Meng et al., 2024), as well as online algorithms such as iterative DPO (Dong et al., 2024) and XPO (Xie et al., 2024). In addition, a line of work has investigated general preference alignment methods that relax the BT model assumption (Azar et al., 2024; Munos et al., 2023; Ye et al., 2024; Wu et al., 2024; Zhang et al., 2024; 2025b). Recently, reinforcement learning algorithms have also achieved notable success in enhancing the reasoning performance of LLMs (Jaech et al., 2024; Guo et al., 2025). A representative example is group relative policy optimization (GRPO) (Shao et al., 2024), which samples a group of responses for each prompt and computes normalized advantages within the group. Several variants of the GRPO objective (Liu et al., 2025; Zheng et al., 2025) have been proposed, differing in how they compute the importance ratio or the advantage. However, a key limitation of GRPO is that when all responses within a group are either correct or incorrect, the advantages collapse to zero, yielding no gradient for training. To address this issue, DAPO (Yu et al., 2025) introduces a dynamic sampling strategy that repeatedly samples new prompts and responses until the batch contains a sufficient number of prompts with non-zero reward variance. But this sampling strategy introduce significantly high computation costs for response generation, which becomes computation bottleneck for larger models. In this paper, we propose AR3PO, an algorithm that mitigates the vanishing advantage issue in a more sampling efficient manner and achieves performance comparable to DAPO while requiring significantly lower generation costs.

**Adaptive Rollout.** Recent works have explored adaptive rollout strategies for GRPO and related algorithms. Liao et al. (2025) allocate rollout budgets based on the ranking of prompts by their historical average reward, giving more budget to difficult prompts to encourage correct responses. Yao et al. (2025) estimate the

required number of rollout samples to minimize gradient variance in the optimization objective. A concurrent work (Yang et al., 2025) also designs a rollout allocation method which uses the cumulative accuracy to calculate a weighting function. The key distinction is that these methods focus on developing more effective allocation strategies to improve the performance, whereas our approach adopts a multi-stage strategy that aims to generate useful training signals more efficiently, thereby reducing generation costs.

**Rollout Replay.** The idea of rollout replay is well established in reinforcement learning and can be traced back to the experience replay technique in Deep Q-Networks (DQN) (Mnih et al., 2015), where past transitions are stored and repeatedly sampled to update the Q-network. More recently, several concurrent works have applied rollout replay techniques in the RLVR setting to improve reasoning performance. Sun et al. (2025) directly utilize recent rollout responses from the replay buffer to update the policy. Zhang et al. (2025a) sample previous correct responses for all prompts and combine them with on-policy rollouts during training. In contrast, our method differs in two key aspects: (1) we only reuse previously generated correct responses when no correct response is present in the current group; (2) we introduce two new techniques to mitigate the distribution shift between the current policy and the behavior policy that generated the previous response.

## 6   Conclusion and Future Work

In this paper, we introduced AR3PO, a novel sampling efficient RLVR approach that integrates two techniques: (1) *adaptive rollout*, a dynamic response generation strategy that allocates more responses to difficult prompts while reducing computation on easy ones, and (2) *response reuse*, which leverages previously generated responses to provide useful training signals. Experiments on multiple mathematical reasoning benchmarks with both Qwen and Llama base models demonstrate that AR3PO achieves performance comparable to or better than strong baselines such as GRPO and DAPO, while requiring significantly lower rollout cost. In the future, we plan to extend AR3PO to LLM agent settings and improve the efficiency of trajectory sampling.

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
