# OpenReview forum: "Improving Sampling Efficiency in RLVR through Adaptive Rollout and Response Reuse"
_TMLR — Withdrawn by Authors_

### Review · Reviewer_ysgh · 2026-04-20

**Summary Of Contributions:**

**Summary：**

The paper proposes AR3PO (Adaptive Rollout and Response Reuse Policy Optimization), a more sampling-efficient RLVR method for reasoning-oriented LLM post-training. The key idea is to address the zero-advantage issue in GRPO-style training without incurring the high rollout cost of DAPO. AR3PO combines adaptive rollout, which allocates more responses to harder prompts and fewer to easier ones, with response reuse, which leverages previously generated correct responses to recover training signal when the current rollout contains no correct sample. Experiments on mathematical reasoning benchmarks with Qwen and Llama models show that AR3PO consistently improves over GRPO and remains competitive with DAPO while using substantially fewer sampled responses.

**Key strengths：**
- The paper addresses an important problem in RLVR, namely the inefficiency caused by zero-variance reward groups in GRPO-style training.
- The proposed method is simple and well motivated: adaptive rollout and response reuse directly target two clear sources of wasted sampling and missing training signal.
- The empirical results are solid: AR3PO is consistently stronger than GRPO and achieves similar performance to DAPO with substantially lower rollout cost.

**Key weaknesses：**
- Reusing past correct responses changes the effective training distribution, but the paper does not deeply analyze possible downsides such as overfitting to replayed successful trajectories or reduced exploration diversity.
- While the efficiency gains are compelling, the improvements in final task performance over strong baselines such as DAPO are relatively modest.
- More details：It is not fully clear how sensitive AR3PO is to design choices such as the number of rollout stages, buffer composition, or reuse frequency.
- The efficiency advantage over DAPO is promising, but the comparison is not fully controlled, since the two methods are evaluated under different rollout settings and response length limits. This makes the efficiency claim less conclusive in that section.

**Audience:**

Yes

**Audience Explanation:**

Yes.
I believe at least part of the TMLR audience would be interested in this paper. The work studies a timely topic and focuses on a practically important issue: improving sampling efficiency in GRPO-style reasoning training. This is relevant to researchers working on reinforcement learning for language models, reinforcement fine-tuning, and efficient post-training methods. Even though the evaluation is mostly limited to mathematical reasoning, the paper addresses a broader methodological question about how to allocate rollout budget and reuse past responses more effectively in verifiable-reward settings.

**Claims And Evidence:**

Yes

**Claims Explanation:**

Mostly.

Explanation:
The main claims are generally supported by the empirical evidence. The paper compares AR3PO against the most relevant baselines (GRPO and DAPO), evaluates on multiple math reasoning benchmarks across two model families, and includes useful ablations and training-dynamics analysis. These results support the claim that AR3PO is more sample-efficient than GRPO and remains competitive with DAPO while using fewer sampled responses.

However, the evidence is stronger for sampling efficiency than for broader generality. The evaluation is mostly limited to math reasoning, and the 32B comparison with DAPO is not fully controlled due to differences in rollout settings and response length limits. So while the core claims are convincing, some of the broader efficiency claims should be interpreted with a bit more caution.

**Requested Changes:**

- Critical for acceptance: The paper’s efficiency claim against DAPO in the larger-scale setting should be better controlled or stated more carefully. In Section 4.5, the comparison is made under different rollout settings and response length limits, and the paper also notes that an exact rollout-cost comparison is not possible there. The authors should either add a more matched comparison or explicitly narrow the claim to the settings where the evidence is strongest.
- Would strengthen the work: The paper should provide a more explicit analysis of possible downsides of response reuse, such as overfitting to replayed successful trajectories or reduced exploration diversity. Since response reuse is a central component of the method, a short discussion or empirical probe of these risks would improve the paper.
- Would strengthen the work: The authors should include more detail on the sensitivity of AR3PO to key design choices, such as the number of rollout stages, replay buffer composition, and reuse frequency. This would make the method easier to evaluate and reproduce, and would clarify how robust the reported gains are.
- Would strengthen the work: The paper would benefit from framing its contribution more explicitly as an efficiency improvement rather than a clear accuracy improvement over DAPO, since the final performance differences are relatively modest. A clearer discussion of this tradeoff would better align the claims with the empirical results.

---

### Review · Reviewer_GmGE · 2026-04-23

**Summary Of Contributions:**

The paper presents a novel method of sampling responses for RLVR, called AR3PO. It proposes using dynamic rollouts with less generations for easier prompts and more generation for tougher prompts. Moreover, in case some queries have no correct response, they suggest using a correct answer from previous stages and suggest two different ways of handling this training.

Strengths:
- The paper clearly motivates the requirement of the improved sampling strategy.
- The paper clearly explains the two different methods of reusing prior correct responses.
- The method has the potential to significantly improve sampling efficiency while maintaining performance.

Weaknesses:
- The paper does not clearly explain how the buffer for correct answers is maintained. Is it per query?
- The paper does not explain if there are any additional memory costs associated with maintaining the buffer.
- Most of the sampling efficiency is over DAPO and not that much over GRPO. The main advantage seems to come from having a smaller batch size of the AR3PO. However smaller batch size means more training steps? So it the sampling advantage is not clear.
- Results on 32B model are not presented properly.
- The paper does not compare AR3PO with the baselines mentioned in Adaptive Rollout and Rollout Replay and does not provide a convincing reason for not using them. While they mention what they do differently, it is not clear why these related works are not eligible for being a baseline as they also focus on efficiency of sampling.

**Audience:**

No

**Audience Explanation:**

The paper presents a novel RLVR algorithm which has the potential to significantly improve sampling efficiency while maintaining performance. However, several concerns remain with the sampling efficiency shown in experiments. The authors show the sampling efficiency per batch which is affected by the high batch size of DAPO. Instead they should show it with responses generated in the whole training process as higher batch size can mean less training steps. Another alternative could be fixing the total number of response generations in the training process and comparing the performances.

**Claims And Evidence:**

No

**Claims Explanation:**

- The paper does not clearly explain how the buffer for correct answers is maintained. Is it per query?
- The paper does not explain if there are any additional memory costs associated with maintaining the buffer.
- Most of the sampling efficiency is over DAPO and not that much over GRPO. The main advantage seems to come from having a smaller batch size of the AR3PO. However smaller batch size means more training steps? So it the sampling advantage is not clear.
- Results on 32B model are not presented properly.
- The paper does not compare AR3PO with the baselines mentioned in Adaptive Rollout and Rollout Replay and does not provide a convincing reason for not using them. While they mention what they do differently, it is not clear why these related works are not eligible for being a baseline as they also focus on efficiency of sampling.

**Requested Changes:**

- Explain clearly if the Buffer is maintained per query and if there are additional memory requirements. If so, how much additional memory is required over GRPO/DAPO.
- In section 3.2, Og is never defined. Please define it.
- In section 3.2, option 1 of response reuse, it is mentioned replace the denominator of rc,t(θ′) with πθ(oc,t |x,oc,<t). Is this the reference model? What is the difference from the original formulation. It is not clear.
- In section 4.1, it is mentioned that GRPO and DAPO uses 8 responses and A3RPO uses 4 in 2 stages. If I understand correctly, this means DAPO always generates 8 responses. But isn't DAPO a dynamic algorithm? Please explain this clearly.
- Please report the performance and sampling efficiency of GRPO and DAPo with 4 rollouts instead of 8 comparing it with 4x2 for A3RPO.
- Please report the performance and sampling efficiency of DAPO with 512 batch size as the main improvement seems to come from using smaller batch size.
- Please connect the sampling efficiency with total rollouts in the whole generation process and not per batch.
- Add an additional experiment showing performance of GRPO, DAPO and AR3PO when the total number responses (budget) is fixed for the training process.
- Provide results table with GRPO, DAPO and AR3PO similar to how it is shown for Qwen-7B.
- Clarify why the works mentioned in related works (Adaptive Rollout and Rollout Replay) do not qualify for being baselines. While it is clear that AR3PO is different from them, it is not clear what they lacked and why they do not qualify to be shown as baselines alongside GRPO and DAPO.

---

### Review · Reviewer_A8j7 · 2026-04-23

**Summary Of Contributions:**

The paper addresses a critical issue in GRPO, where the advantage can vanish during training. To mitigate this, the authors propose AR3PO, which incorporates adaptive rollout and response reuse to improve sample efficiency and maintain consistent training signals. Experimental results demonstrate that the proposed method outperforms existing RLVR approaches, such as GRPO and DAPO, in both accuracy and efficiency.

**Audience:**

No

**Audience Explanation:**

The positive aspect of this paper is that it addresses an important and cutting-edge research question in RLHF for building reasoning models.

However, the core research question—the advantage vanishing problem—appears to overlap with issues previously studied in PPO and subsequent work. The authors should clarify how their approach avoids revisiting the same fundamental problem, and more importantly, how the proposed method differs from or outperforms conventional strategies that rely on value functions or value networks.

**Broader Impact Concerns:**

No ethnical concerns.

**Claims And Evidence:**

No

**Claims Explanation:**

I think the main issue is that, while the motivation of resolving vanishing advantages is reasonable when compared to GRPO, it is unclear how this differs from PPO. In PPO, a value network is used to estimate advantages. In other words, how does the proposed method outperform PPO at a theoretical level? For example, a drawback illustrated in Figure 1 of DAPO is that it loops back to the original question, requiring multiple additional prompts. We need to ensure that resolving vanishing advantages does not reintroduce the original motivation behind PPO.

I also have several questions:

- In GRPO, the issue where all responses are either correct or incorrect may potentially be mitigated by increasing \( G \). Could you compare your method against this approach?

- Since \( k \) responses are generated for each query, does this not introduce additional computational cost? Could you briefly explain how the performance gain (i.e., speedup) reported in Table 1 is achieved, perhaps from a complexity perspective?

- It is still possible that no correct answer is obtained before reaching the maximum generation steps. In such cases, how does response reuse help if no correct responses exist? How do you ensure robustness against this scenario?

- Is a verifier used to determine correctness? If so, how does this differ from using a value network to estimate values or advantages?

- Could you elaborate on how hyperparameters are selected? For key parameters such as \( S \) and \( k \), do you perform hyperparameter sweeps? If so, it would be helpful to include these results (e.g., in the Appendix).

- Mathematical benchmarks allow straightforward correctness verification. How does your method extend to tasks such as IMOBench or other settings requiring LLM-based judges (e.g., creativity tasks or agent benchmarks like PinchBench)? In these cases, where evaluation is score-based, how is correctness defined? Additionally, these datasets may not be sufficiently challenging for models like Qwen 2.7B. You may consider evaluating on more difficult benchmarks.

- Could you also compare your RL-tuned models with industrial-scale reasoning models such as DeepSeek-R1-Qwen, Nemotron, or DeepSeek-R1-LLaMA? If possible, including experiments with more recent base models and their corresponding RL-tuned versions would further strengthen the paper.

**Requested Changes:**

In alignment with the concerns above, I believe the following aspects should be addressed:

1. **Clarify the motivation** and provide a precise definition of values and rewards, including how advantages are computed and interpreted in this framework.

2. **Discuss failure cases**, particularly scenarios in which no correct answer is ever obtained. How does the method behave in such cases, and how is training stability ensured?

3. **(Appendix)** Provide a systematic explanation of how hyperparameters are selected (not just the final choices). For example, include search ranges, sensitivity analyses, or ablation studies.

4. **Evaluate on additional reasoning benchmarks**, especially those involving open-ended or less verifiable questions, to demonstrate generalizability beyond deterministic tasks.

5. **Include comparisons with industry-grade reasoning models**, to better contextualize performance and demonstrate practical relevance.

---

### Note · Authors · 2026-05-25

I have read and agree with the venue's withdrawal policy on behalf of myself and my co-authors.